# Promising Immunotherapeutic Modalities for B-Cell Lymphoproliferative Disorders

**DOI:** 10.3390/ijms222111470

**Published:** 2021-10-25

**Authors:** Jana Mihályová, Katarína Hradská, Tomáš Jelínek, Benjamin Motais, Piotr Celichowski, Roman Hájek

**Affiliations:** 1Department of Haematooncology, University Hospital Ostrava, 708 52 Ostrava, Czech Republic; katarina.hradska@fno.cz (K.H.); tomas.jelinek@fno.cz (T.J.); roman.hajek@fno.cz (R.H.); 2Faculty of Medicine, University of Ostrava, 708 52 Ostrava, Czech Republic; benjamin.motais@fno.cz (B.M.); piotr.celichowski@fno.cz (P.C.)

**Keywords:** immunotherapy, bispecific antibodies, antibody-drug conjugates, brentuximab vedotin, polatuzumab vedotin, mosenutuzumab, epcoritamab, glofitamab

## Abstract

Over the last few years, treatment principles have been changed towards more targeted therapy for many B-cell lymphoma subtypes and in chronic lymphocytic leukemia (CLL). Immunotherapeutic modalities, namely monoclonal antibodies (mAbs), bispecific antibodies (bsAbs), antibody-drug conjugates (ADCs), and chimeric antigen receptor T (CAR-T) cell therapy, commonly use B-cell-associated antigens (CD19, CD20, CD22, and CD79b) as one of their targets. T-cell engagers (TCEs), a subclass of bsAbs, work on a similar mechanism as CAR-T cell therapy without the need of previous T-cell manipulation. Currently, several anti-CD20xCD3 TCEs have demonstrated promising efficacy across different lymphoma subtypes with slightly better outcomes in the indolent subset. Anti-CD19xCD3 TCEs are being developed as well but only blinatumomab has been evaluated in clinical trials yet. The results are not so impressive as those with anti-CD19 CAR-T cell therapy. Antibody-drug conjugates targeting different B-cell antigens (CD30, CD79b, CD19) seem to be effective in combination with mAbs, standard chemoimmunotherapy, or immune checkpoint inhibitors. Further investigation will show whether immunotherapy alone or in combinatory regimens has potential to replace chemotherapeutic agents from the first line treatment.

## 1. Introduction

In the last decade, immunotherapy experienced a very rapid development and established itself as one of the fundamental parts of cancer treatment. Its main aim is to activate and encourage the body’s own immune system to distinguish and destroy malignant cells [1]. In 1997, rituximab was the very first CD20-specific mAb to obtain regulatory approval, and rapidly became a key part of chemotherapeutic regimens (so-called chemoimmunotherapy) in the care for both B-cell non-Hodgkin lymphoma (B-NHL) and chronic lymphocytic leukaemia (CLL) [2]. Since then, antibody drug-conjugates (ADCs), immune checkpoint inhibitors, and both bispecific and trispecific antibodies for CAR-T cell therapy have been developed. Each class of these immunotherapeutic modalities has a unique mechanism of action, structure, and pharmacokinetic properties which are reflected in their specific treatment-related toxicities.

Mature B-cell neoplasms are a diverse group of lymphoid disorders with varying clinical manifestations, pathologic features, and outcomes. Diffuse large B-cell lymphoma (DLBCL) along with follicular lymphoma (FL) represent the most common aggressive and indolent histological subgroups of B-NHL with the incidence of ~5.6/100.000/year and ~2.7/100.000/year respectively (https://seer.cancer.gov/statfacts/html/dlbcl.html accessed on 19 October 2021). Together they comprise 50% of all B-NHL cases [3]. Chronic lymphocytic leukemia is another indolent B-cell malignancy with an incidence of ~4.9/100.000/year (https://seer.cancer.gov/statfacts/html/dlbcl.html accessed on 19 October 2021), and accounts for 25–30% of newly diagnosed leukemias in the Western world [4]. Expression of tumor cell antigens varies according to diagnosis and may change during the treatment. Nevertheless, some markers such as B-cell-associated antigens (CD19, CD20, CD22, and CD79b) are commonly expressed on mature B-lymphocytes including neoplastic ones, and became targets for different immunotherapeutic modalities [5]. Despite that, molecules more selectively expressed on malignant cells and ideal targets on effector cells (e.g., T-cells, NK cells, APCs—antigen presenting cells, macrophages) are still being explored. Some of such antigens (CD30, CD47, CD37, CD70) will be described later in more detail.

Our review is focused on the current role of ADCs and bsAbs in B-NHL and CLL treatment. We discuss mechanism of action specific for each class, point to differences in their serious treatment-related toxicities and give an overview of the most important preclinical and clinical trials. Figure 1 shows examples of some immunotherapeutic agents used in B-NHLs.

## 2. Antibody-Drug Conjugates

Antibody-drug conjugates could be considered a bridge between conventional chemotherapy which eliminates all rapidly dividing cells and more targeted immunotherapy combining the best anti-tumor properties of each of them [6].

Antibody-drug conjugates consist of three main parts: the mAb, the cytotoxic payload, and the linker, which connects the two together. Monoclonal antibody serves as a transporter of the payload and binds to the specific antigen on the surface of tumor cells. The antigen then triggers the internalization of the whole complex via endocytosis, the linker is degraded by lysosomal enzymes and the payload is released. Upon release, the cytotoxic agent eliminates the tumor cell by various mechanisms, most often by DNA damage or interaction with microtubules [7,8]. Several ADCs (e.g., brentuximab vedotin) exert their anti-tumor toxicity also via bystander effect that is characterized as capability of the payload to permeate through cell membranes and kill neighboring cells, regardless of the presence of the target antigen [6].

Monoclonal antibody should be directed against antigen highly expressed on tumor cells and with limited or no expression on normal tissues to avoid excessive off target toxicity [7]. Payload is required to be highly toxic in subnanomolar concentrations, to be able to conjugate with the antibody, and to remain stable in physiological conditions [6,8]. Nevertheless, the essential components of ADC from the perspective of biochemical stability are linkers and there are two types of them (cleavable and non-cleavable) [9]. The first group has the cleavage site located between payload and monoclonal antibody and the “separation” process may occur within the endosome, lysosome or cytosol itself, while the second group needs complete proteolytic degradation of mAb backbone within the lysosome to release the payload [7,8,9]. Another important attribute of ADC in terms of efficacy and safety is drug-antibody ratio (DAR) which is defined by the number of cytotoxic molecules attached to mAb and relates to ADC’s homogeneity and conjugation strategies [7,8,10].

As of 2021, eleven ADCs have been approved by the US Food and Drug Administration (FDA) in various hematological and oncological indications (Table 1) [8,10,11]. And many other ADCs are undergoing various stages of clinical testing. Results of phase I/II clinical trials, summary of currently ongoing phase III clinical trials and chemotherapy free clinical trials of ADCs are to be found in Table 2 and Table 3.

### 2.1. Antibody-Drug Conjugates Approved for the Treatment of B-Cell Non-Hodgkin Lymphomas

#### 2.1.1. Polatuzumab Vedotin, POLIVY^TM^, DCDS4501A, RG7596 (Roche)

Polatuzumab vedotin (PV) consists of a humanized IgG1 mAb against CD79b (B-cell receptor component) and a microtubule-disrupting agent MMAE (monomethyl auristatin E) conjugated by a protease-cleavable linker [28,29,30].

GO29365 (NCT02257567) was an open-label, international, multi-center, randomized, phase Ib/II clinical trial evaluating safety and efficacy of PV plus bendamustine and rituximab (BR) or bendamustine and obinutuzumab (BG) in relapsed/refractory (RR) DLBCL and FL setting compared to BR alone [12,30]. The results proved superiority of PV plus BR opposed to BR. Overall response rate (ORR) was 45% (18/40) versus 18% (7/40), complete response (CR) was 40% (16/40) versus 18% (7/40), duration of response (DOR) was 10.3 versus 4.1 months, median progression free survival (PFS) was 7.6 versus 2.0 months and overall survival (OS) was 12.4 versus 4.7 months in RR DLBCL, but there was no benefit in RR FL patients [12,29]. Based on the results of this study, PV in combination with BR was approved for the treatment of transplant ineligible RR DLBCL patients after at least one (Europe) or two (USA) prior lines of therapy in combination with BR in 2019 [31].

Polatuzumab vedotin in combination with CHP (cyclophosphamide, doxorubicin, prednisone) and rituximab (R) or obinutuzumab (G) were also investigated in treatment naïve DLBCL patients in a phase I/II clinical trial (NCT01992653) [13]. Sixty-six patients were included in the phase II part. Overall response rate was 89% (59/66) with CR of 77% (51/66). 12-month PFS and OS were 91% and 94%, respectively. There was no significant difference in efficacy between PV plus R-CHP or PV plus G-CHP cohorts. Based on these promising results, a phase III study comparing standard of care for newly diagnosed (ND) DLBCL, R-CHOP (rituximab, cyclophosphamide, doxorubicin, vincristine, prednisone), with PV-R-CHP (NCT03274492, Polarix) has been ongoing and the results are impatiently awaited.

Other PV combination investigated in RR DLBCL with available primary data is PV plus rituximab and lenalidomide [14]. DLBCL cohort in this phase Ib/II clinical trial (GO29834, NCT02600897) was characterized by the median age of 71 years (28–92) and median of 2 prior therapies. The investigator-assessed best overall response was 74% (36/49) with 35% (17/49) of CR. It is worth mentioning that CRs were durable with 82% (14/17) of patients remaining in remission at the cutoff date which is a promising sign in this difficult-to-treat population. The same clinical trial included also a RR FL arm investigating PV plus obinutuzumab and lenalidomide and the interim results were presented at British Society for Haematology 2020 Virtual meeting [15]. Efficacy population included 56 highly pretreated patients (median of three prior lines of therapy). At the median follow-up of 15.1 months, the data are still immature, but investigator-assessed ORR was 83% (38/46) with CR of 61% (28/46) by Modified Lugano 2014 criteria. Twelve-month PFS was 83.4%. This triplet appears to be a potent option for RR FL patients. However, longer follow-up is warranted for either arm.

Preclinical data suggest synergistic anti-tumor effect of PV and venetoclax (inhibitor of anti-apoptotic Bcl-2 protein) when PV promotes MCL-1 (myeloid leukemia cell 1) degradation, a known mechanism of resistance to venetoclax (Ven) [32]. That was the rationale of a phase Ib/II study (GO29833, NCT02611323) investigating the combination of PV with Ven and G for RR FL patients [33]. Study population was highly pretreated (74% ≥ 2 prior therapies) and 55% of patients were double refractory (to anti-CD20 mAb and alkylating agent). Primary analysis was presented at ASCO (American Society of Clinical Oncology) Meeting 2021. At the end of induction, PET (positron emission tomography)-CR was 57% (28/49). Median PFS was not reached and 12-month PFS was 73% with a median follow up of 14.4 months [33]. These results look encouraging, but data from maintenance treatment and beyond will be crucial to determine the benefit for this patient population.

Another promising PV combination for RR B-NHL patients is with mosunetuzumab (bispecific antibody targeting CD20 and CD3) and will be discussed further in text.

#### 2.1.2. Loncastuximab Tesirine, ZYNLONTA^TM^, ADCT-402 (ADC Therapeutics S.A.)

Loncastuximab tesirine (lonca) comprises of a humanized anti-CD19 IgG1 mAb attached to DNA-damaging agent, pyrrolobenzodiazepine (PBD) dimer toxin, by a protease-cleavable linker [7,17].

In 2021, Hamadani et al. published final results of a phase I study investigating this ADC as a monotherapy in RR B-NHL (NCT02669017) [17]. Patients had received median of three prior lines of systemic therapy. Overall response rate was 45.6% (82/180) in all evaluable patients with CR of 26.7% (48/180). Median DOR (duration of response) was 5.4 months for the whole cohort and was not reached for DLBCL patients who achieved CR. As this was primarily the dose-escalation and dose-expansion study, recommended dose for a phase II trial (LOTIS-2, NCT03589469) aiming at RR DLBCL was determined as 150μg/kg every three weeks for two doses followed by 75μg/kg every three weeks [16]. Interim results showed ORR 48.3% (70/145) with 24.8% CR (36/145). Median DOR for responders (CR + PR; PR = partial remission) was 12.6 months and not reached for patients with CR. Median PFS and median OS were 4.9 and 9.5 months, respectively. Lonca showed efficacy even in high risk groups such as double or triple-hit, transformed or refractory DLBCL. These encouraging results led to the FDA approval of lonca monotherapy for large B-cell lymphomas (DLBCL, transformed DLBCL, high grade B-cell lymphoma) after at least 2 prior lines of therapy in April 2021 [34].

Besides monotherapy, lonca is currently being tested in combination with ibrutinib in RR DLBCL and RR MCL in phase I/II trial (LOTIS-3, NCT03684694). Preliminary results for 36 evaluable patients from a phase I part were recently presented at ICML (International Conference on Malignant Lymphoma) 2021 and the efficacy data looked promising with ORR of 63.9% (23/36) and CR of 36.1% (13/36) [18]. Another investigated combination includes lonca plus rituximab versus R-GemOx (rituximab, gemcitabine, oxaliplatin) for RR DLBCL patients in a phase III study (LOTIS-5; NCT04384484), but the results are not available yet [35]. Loncastuximab tesirine is also being considered for the treatment of RR FL in a phase II clinical trial (LOTIS-6; NCT04699461) opposed to idelalisib [36].

### 2.2. Other Antibody-Drug Conjugates Investigated in B-Cell Non-Hodgkin Lymphomas

#### 2.2.1. Brentuximab Vedotin, ADCETRIS^®^, SGN-35 (Seagen/Takeda Oncology)

Brentuximab vedotin (BV) consists of a microtubule-disrupting agent MMAE, but this time covalently attached by a protease-cleavable linker to the humanized mAb directed against CD30 [37,38]. Transmembrane receptor CD30 can be found on classical Hodgkin lymphoma´s (cHL) Reed-Sternberg cells, on some T-cell malignancies and on a subset of DLBCL as well [28,38]. As of 2021, BV has been approved for the treatment of RR cHL, ND, and RR CD30+ systemic anaplastic large cell lymphoma and RR CD30+ cutaneous T-cell lymphoma [37,39,40,41,42].

Primary mediastinal large B-cell lymphoma (PMBCL) is a specific subtype of DLBCL which shares some characteristics with cHL, especially high CD30 expression [43]. That is why similar efficacy was expected from BV monotherapy in RR PMBCL patients. A phase II study investigating BV in this setting (NCT02423291) did not confirm this hypothesis with ORR of only 13.3% (2/15) and with no CR achieved. Therefore, BV alone is considered inactive in RR PMCBL. Nevertheless, Zinzani et al. presented the data of a phase I/II combination trial of BV with nivolumab (immune checkpoint inhibitor) in RR PMBL at ICML 2021 and the results were just the opposite (NCT02581631, CheckMate436) [44]. Thirty highly pretreated patients were recruited with a median of two prior lines of therapy. Results from the extended median follow up of 33.7 months showed ORR of 73% (22/30) with CR of 37% (11/30). Median DOR was 31.6 months and median duration of CR and median OS have not been reached. As this combination proved its efficacy and long-term survival benefits, a phase II study of BV + nivolumab with R-CHP (rituximab, cyclophosphamide, doxorubicin, prednisone) for newly diagnosed ND PMBCL is currently underway (PACIFIC, NCT04745949).

Not only PMBCL, but CD30+ DLBCL and gray zone lymphoma (GZL) could be appropriate targets for BV combination therapy. In a phase 1/2 multicenter trial (NCT01994850), 29 patients with ND CD30+ B-cell lymphomas (22 PMBCL, 5 DLBCL, 2 GZL) were treated with 6 cycles of BV + R-CHP and 52% of patients followed consolidative radiotherapy [19]. Overall response rate was 100% with 86% CR and there was no difference between radiotherapy + or-cohort. At the median follow-up of 30 months, 2-year PFS and OS were 85% and 100%, respectively. These results are quite promising but should be regarded with caution due to the small study cohort.

At ICML 2021, preliminary results from a phase I combination trial of BV plus lenalidomide and rituximab for RR DLBCL were foreshadowed [45]. Among 37 evaluated patients, ORR was 56.7% (21/37) and median DOR was 13.2 months for the responding patients (CR+PR). The PFS and median OS reached 11.2 months and 14.3 months, respectively. It is worth mentioning that the results were similar between CD30+ and CD30 < 1% cohorts. Based on these promising outcomes, a phase III study comparing lenalidomide and rituximab plus BV/placebo in RR DLBCL setting was initiated (ECHELON-3, NCT04404283).

#### 2.2.2. Naratuximab Emtansine, Debio1562, IMGN529 (Debiopharm)

Naratuximab emtansine (nara) represents a humanized anti-CD37 IgG1 antibody conjugated via a non-reducible thioether linker to a potent anti-mitotic agent—maytansinoid DM1 [7,46] Transmembrane protein CD37 has the highest abundance on mature B-cells and it is expressed only in a low level on T-cells and myeloid cells [7,47]. Under physiological condition, CD37 plays a role in normal B-cell activation and survival. On the other hand, it is highly expressed on most histological subtypes of B-NHL.

Naratuximab emtansine was evaluated in a phase I study (NCT01534715) in heavily pretreated RR NHL patients (median of three prior systemic regimens) and the results were encouraging with ORR 22% (4/18) in DLBCL cohort [46]. Based on these results, nara was selected for a phase II clinical trial in combination with rituximab for RR B-cell NHL patients (NCT02564744) [48]. Seventy-four RR DLBCL patients were evaluable for efficacy and ORR of 43.2% (32/74) with 32.4% (24/74) of CRs were observed in this group. Median DOR was not reached during median follow-up in responders of 13.7 months.

#### 2.2.3. Other Antibody-Drug Conjugates

There are several ADCs that were tested in B-NHL setting but were not pursued further due to safety or efficacy reasons. Pinatuzumab vedotin (anti-CD22 mAb + MMAE) was initially investigated together with PV in a phase II study ROMULUS, but was abandoned thereafter for PV´s better results [49]. Camidanlumab tesirine (anti-CD25 mAb + PBD) is currently more extensively researched in cHL than B-NHL setting [50,51]. Coltuximab ravtansine (anti-CD19 mAb + microtubule disruptive agent DM4) showed only a moderate efficacy in RR DLBCL in contrast with another CD19 targeting agent—lonca [52,53]. Three CD70 targeting ADCs (SGN-CD70A, MDX-1203, vorsetuzumab mafodotin) raised safety concerns due to treatment-emergent toxicities [7,54,55,56]

## 3. Bispecific Antibodies

Bispecific antibodies (bsAbs) are immunotherapeutic agents designed to recognize and bind two different antigens or two different epitopes on the same antigen. Many formats have been produced so far, but in general, bsAbs can be divided into two major classes, those bearing an Fc region and those lacking an Fc region. The former resembles IgG or IgG-like molecules with Fc-mediated effector functions (ADCC—antibody-dependent cell cytotoxicity, CDC—complement-dependent cytotoxicity, ADCP—antibody-dependent cell phagocytosis) and improved stability [57]. The latter is represented by small molecules composed of either a complete fragment antigen-binding (Fab) region or only some parts of Fab immunoglobulin fragments connected with a peptide linker [57]. Small molecules gain an advantage of better tissue penetration, but on the other hand, they have no carrier system protecting them from rapid intracellular degradation and renal elimination. Due to their properties, many of these bsAbs require frequent or even continual administration [57,58,59].

The mechanism of action depends on biological targets of particular bsAb. Currently, T-cell engagers (TCEs) that typically target tumor associated antigen (TAA) with one arm, and T-cell binding domain with another one, represent the most frequently used class of bsAbs. Protein CD3 is a part of T-cell receptor (TCR) signaling complex [60], and acts as the binding domain of many effector T-cells. In contrast to physiologically activated T-lymphocytes, immunological synapse mediated via TCEs induces polyclonal T-cell activation independent of TCR epitope specificity, independent of major histocompatibility complex (MHC), and co-stimulatory signals [60]. Once activated, cytotoxic lymphocytes release granules containing perforins and granzymes and mediate lysis of the target cells [58,61,62]. Afterwards, multiple cytokines such as interleukin (IL)-2, IL-6, IL-10, interferon-gamma (IFN-γ), or tumor necrosis factor-alpha (TNF-α) are secreted and, besides T-lymphocytes, other immune cells are activated (e.g., B-cells, macrophages, NK cells, etc.) [58,61].

To optimize the way of immune system activation, alternative antigens on different effector cells (e.g., CD137 also known as 4-1BB on T-cells; CD16 on NK cells; CD40 on APCs) including inhibitory immune checkpoints or their ligands (A2AR/adenosine, CTLA-4/CD80, and CD86, KIR/MHC class I, LAG3/MHC class II, PD-1/PD-L1 and PD-L2) are still being explored [63,64].

### 3.1. Bispecific Antibodies Anti-CD20xCD3

Results of phase I/II clinical trials and summary of currently ongoing clinical trials of anti-CD20xCD3 TCEs are shown in Table 4 and Table 5.

#### 3.1.1. Mosunetuzumab, RG7828 (Roche)

Mosunetuzumab (M) is a full-length, fully humanized IgG1 TCE targeting CD20 and CD3. In a phase I/II dose-escalation and expansion study (NCT02500407), mosunetuzumab administrated intravenously (i.v.) (8–17 cycles) was evaluated in patients with RR B-NHL (median of 3 prior therapies, 23 patients were previously treated with CAR-T cell). Across all dose levels, 34.7% (41/119) of patients responded to treatment and 18.6% (22/119) achieved CR in a group of aggressive lymphoma (DLBCL and transformed FL) [20]. Slightly better outcomes were observed in a group of RR FL with the ORR and CR demonstrated in 68% (42/62) and 50% (31/62), respectively. The median PFS for FL patients was 11.8 months (95% CI: 7.3–21.9 months) [65]. Based on these results, FDA granted Breakthrough Therapy Designation for FL patients who have received at least two prior systemic therapies. Within the same study, mosunetuzumab administrated subcutaneously (s.c.) showed ORR and CR in 86% (6/7) and 29% (2/7) of indolent B-NHL, and 60% (9/15) and 20% (3/15) of aggressive B-NHL. With median follow-up of 6.9 months, median PFS was not reached in both groups [21]. Treatment was moved to the first line for elderly or unfit DLBCL patients not eligible for full-dose chemoimmunotherapy (ORR 67.7% (21/31), CR 41,9% (13/31)) [22].

Early clinical data for a phase I/II study (NCT03671018) of mosenutuzumab plus PV administrated i.v. (8–17 cycles of mosenutuzumab plus 6 cycles of PV) were presented at European Hematology Asociation (EHA) congress 2021. In a dose escalation cohort, 68% (15/22) of all RR B-NHL (DLBCL and FL with a median of three previous therapies) responded to treatment and 54,5% (12/22) achieved CR. [23]. The phase expansion cohort of mosenutuzumab plus PV for RR DLBCL (NCT03671018), and phase III clinical trial of mosenutuzumab plus lenalidomide versus R plus lenalidomide for RR FL (NCT04712097) are currently ongoing [66].

#### 3.1.2. Glofitamab, CD20-TCB, RG6026, RO7082859 (Roche)

Glofitamab is a full-length, fully humanized IgG1 TCE with a 2:1 molecular configuration comprising two Fab fragments that bind CD20 and one that binds CD3. In preclinical studies, glofitamab had superior potency than other TCEs with 1:1 format (Bacac, et al. Clin Cancer Res 2018).

In a dose-escalation and dose-expansion phase I clinical trial (NCT03075696) with RR B-NHL, glofitamab (1–12 cycles administrated i.v.) was assessed as a single agent or in combination with obinutuzumab (G). To reduce the risk of CRS, obinutuzumab was administered seven days prior to glofitamab in both arms [67]. Updated data of glofitamab monotherapy showed high response rates across different lymphoma subtypes (median of three previous therapies) with the best outcomes in the highest cohort. For patients with aggressive B-NHL (DLBCL, transformed FL, PMBCL, MCL, and Richter’s transformation), ORR and CR were 48.0% (61/127) and 33.1% (42/127), respectively, and at the recommended dose (step-up dose form 2.5mg to 30mg) for phase II, the ORR and CR were 71.4% (10/14) and 64.3% (9/14), respectively [24]. Patients with indolent B-NHL responded in 70.5% (31/44) and 47.7% (21/44) reached CR [24]. Median PFS was 2.9 (95% CI, 2.1 to 3.9) months, with a plateau of approximately 24% from eight months onward (maximum follow-up of 30 months) in aggressive B-NHL and 11.8 months (95% CI, 6.3 to 24.2) in indolent B-NHL [24].

In a dose-escalation cohort of glofitamab plus obinutuzumab (12 cycles of obinutuzumab plus 1–12 cycles of glofitamab i.v. in a dose ranging from 0.6 to 16mg) which was investigated in various RR B-NHL subtypes (median of two previous therapies), the ORR and CR were 38% (6/16) and 31% (5/16) in aggressive B-NHL and 80% (4/5) (all CR) in indolent B-NHL. Outcomes were even better in a glofitamab target dose of 16 mg (CR 71% (5/7) in aggressive lymphoma, CR 100% (3/3) in indolent lmymphoma) [25].

#### 3.1.3. Epcoritamab, GEN3013, (Genmab, Abbvie)

Epcoritamab is a product of DuoBody technology that generates a full-length anti-CD20xCD3 IgG1 TCE. The bispecific construct is generated by the controlled Fab-arm exchange of two parental IgG1 mAbs (each with a matched single point mutation in the Fc region), and as a result, it can target two different epitopes as a single mAb [68].

A phase I clinical trial (NCT03625037) for aggressive and indolent RR B-NHL (DLBCL, high-grade B-cell lymphoma, MCL, MZL, lymphoma from small lymphocytes) has been already conducted [69] Overall, 46 patients with RR DLBCL (median of three previous therapies) received epcoritamab monotherapy (s.c.) In the group of targeted dose of 48mg, the ORR, CR and PR were 91% (10/11), 55% (6/11) and 36% (4/11), respectively. Four patients who relapsed after CAR-T cell therapy responded (2 CR, 2 PR) to treatment. With a median follow-up of 8.8 months, median PFS was not reached. Efficacy was similar in a group of RR FL (median of 4.5 previous therapies) with ther ORR 80% (10/12), CR 60% (7/12), and PR 20% (2/12) [26]. Currently, a phase III clinical trial comparing epcoritamab monotherapy to standard chemoimmunotherapy in RR DLBCL is ongoing (NCT04628494).

#### 3.1.4. Odronextamab, REGN1979 (Regeneron)

Odronextamab is a fully human IgG4 anti-CD20xCD3 TCE with attenuated Fc region-mediated activity [27]. Heavily pretreated patients with RR B-NHL (median of 3 previous therapies including 12 patients treated with CAR-T cell) received odronextamab (24 doses of odronextamab i.v. in 9 months) in a dose-excalation phase I clinical trial (NCT03888105). Overall, 93% (13/14) of indolent lymphoma patients (FL, MZL, WM) responded to treatment and 71,4% (10/14) achieved CR. Less effective was treatment in the subgroup of aggressive lymphomas (DLBCL, MCL) with the ORR 33% (15/45), CR 18% (8/45) and PR 16% (7/45). Two out of three patients who relapsed after CAR T-cell therapy reached CR [27]. Based on these results, a phase 2 study was designed to assess odronextamab monotherapy given until disease progression in five RR B-NHL cohorts (FL, DLBCL, MCL, MZL and others) [70].

Another phase I ongoing clinical trial (NCT02651662) evaluates combination of odronextamab with cemiplimab (anti-PD-1 immune checkpoint inhibitor) in RR B-NHL (https://clinicaltrials.gov/ accessed on 19 October 2021).

#### 3.1.5. Plamotamab, XmAb13676 (Xencor)

Plamotamab is a humanized IgG1 anti-CD20xCD3 TCE with a bispecific Fc domain that confers its long circulating half-life and stability. In the first-in-human dose-escalating phase I clinical trial (NCT02924402), 42% (15/36) of patients with RR B-NHL (median of 3.5 previous therapies) and 20% (1/5) of patients with RR CLL (median of 4.5 previous therapies) responded to plamotamab monotherapy [71].

#### 3.1.6. IGM-2323 (IGM Biosciences)

IGM-2323 is a bispecific IgM antibody that has ten high-affinity binding domains for CD20 and one for CD3 [72]. Compared with other TCEs, it has a slightly different mechanism of action based on repeatable T-cell activation and high CDC. In preclinical in vitro studies, IGM-2323 irreversibly binds to CD20 positive cells and depletes them with limited cytokine secretion. Nevertheless, so far only limited preliminary data of efficacy and tolerability have been shown in the first-in-human clinical trial with eight RR B-NHL patients [72].

### 3.2. Bispecific Antibodies Anti-CD19xCD3

#### 3.2.1. Blinatumomab, AMG103, MT103 (Amgen)

Blinatumomab was the first in class bispecific T-cell engager (BiTE) construct consisting of two flexibly linked single-chain variable fragments that bind to CD3 and CD19. Molecular weight of blinatumomab is approximately 55 kDa which is reflected in its high systemic clearance and short half-life (2.10 h) [59]. To minimize the risk of treatment-related AEs and maintain plasma concentration, blinatumomab is administrated as a dose-escalating continuous infusion (i.v.) usually lasting four to eight weeks [59,73].

The clinical proof of blinatumomab efficacy was initially demonstrated in patients with RR B-NHL (median of four previous therapies) in the phase I study (NCT00274742), which defined 60 mg/m2/day as the maximum tolerated dose (ORR 69% (24/35), CR 37% (13/35)). In a DLBCL subgroup, 55% (6/11) of patients responded to treatment with 36% (4/11) achieving CR [74]. Responses were significantly better among patients who received the targeted dose with median PFS 1.5 years and median OS 5.8 years [75]. Afterward, several phase II clinical trials were designed for RR B-NHL subsets. One of those assesed blinatumomab as a second salvage in aggressive RR B-NHL patients who did not achieve CR with platinum-based regimen. The ORR, CR, and PR were 37% (15/41), 22% (9/41), and 15% (6/41), respectively. Median OS was 11.2 months among all patients. Among those who achieved CR or proceed to autologous stem cell transplantation (ASCT), median OS was not reached and median PFS was 8.4 months [76].

Combination of blinatumomab with lenalidomide was assessed in a phase I study with RR B-NHL (median od 2.5 previous therapies). At a median follow-up of 14.3 months, the ORR and CR were 83% (15/18) and 50% (9/18), and median PFS was 8.3 months. Three patients who achieved response underwent allogeneic stem cell transplantation and remained in remission for 14.2 to 22.3 months thereafter [77].

Blinatumomab was also tested as consolidation therapy post autologous stem cell transplantation (ASCT) in RR DLBCL or transformed FL [78], post standard chemoimmunotherapy in newly diagnosed high-risk DLBCL [79] or as maintenance therapy post allogeneic stem cell transplantation in RR B-NHL patients [80]. It probably has a potential to increase the response rate in some cases but bigger cohorts and longer follow-up are needed.

#### 3.2.2. Other Bispecific Antibodies

TNB-486 (TeneoBio) is a fully human asymmetric IgG4 anti-CD19xCD3 TCE that was designed to bind CD3 antigen with low affinity. Given the modification, TNB-486 demonstrates efficient tumor cell cytotoxicity coupled with reduced cytokine release (IL-2, IFNγ, IL-6, IL-10 and TNF) without the decrease of secreted granzymes and perforins [81]. This might be an advantage if reduced CRS-related toxicity will be proved in clinical trials.

*Different ImmuneOnco Biopharmaceuticals* generates IMM0306, the first antibody receptor recombinant protein (mAb-Trap) targeting CD47 and CD20. Transmembrane protein CD47 is overexpressed by multiple malignant cells and gives macrophages and other phagocytes “do not eat me” signal. The clinical benefit is based on a hypothesis that binding CD47 may block inhibitory signal mediated via this antigen and proceed the process of phagocytosis [82]. IMM0306 will be soon evaluated in a phase I clinical trial in RR B-NHL patients (NCT04746131) and another, but in this case anti-CD47xCD19 bsAb, namely TG-1801 *(TGTherapeutics),* is currently evaluating as a single agent (NCT03804996) or in combination with umblituximab (anti-CD20 mAb) (NCT04806035) in RR B-NHL and RR CLL patients.

Finally, tebotelimab, MGD01 *(MacroGenics),* a dual-affinity re-targeting antibody (DART), was designed to bind PD-1 and LAG-3 to sustain or restore the function of exhausted T cells. There are only preliminary data from a small cohort of RR DLBCL (ORR 27% (3/11), CR 18% (2/11), PR 9% (1/11)) at the moment. Treatment-related AEs were reported in more than half of patients (64.7%) with one grade ≥3 pneumonia [83]. As this is the dual immune checkpoint inhibitor, autoimune treatment related toxicity should be closely monitored.

## 4. Class Specific Toxicity

Treatment-related toxicity of ADCs depends mainly on the cytotoxic payload and its off-target effect. Both BV and PV contain MMAE, which induces mostly neutropenia (up to 58%) and peripheral neuropathy (27–61% in BV; 43–44% in PV) [12,13,14,19,44]. Brentuximab vedotin-induced neuropathy was predominantly graded 1–2 and usually resolved after treatment discontinuation. Ocular toxicity, mostly extracorneal, was associated with DM4-containing coltuximab ravtansine and affected 25% of patients. Eye disorders were grade 1 or 2, typically occurred during initial treatment cycles [53]. None of these cases required a dose modification and median recovery time was 12.5 days. Loncastuximab tesirine´s warhead is a PBD toxin. The most prominent toxicity seen with the use of this ADC was thrombocytopenia (30–71%), neutropenia (40–59%), elevated gamma-glutamyl transferase (GGT) levels (31–41%), fatigue (21–43%) and skin-related toxicity (up to 54%) such as rash, erythema, pruritus or maculopapular rash, especially on sun-exposed areas [16,17,18]. Peripheral edema (32%) and pleural effusion (21%) also raised safety concerns during the clinical testing, but the introduction of dexamethasone as premedication led to notable decrease in frequency of these events [17].

T-cell engagers have encountered hurdles in the clinical practice dominantly related to cytokine release syndrome (CRS) and immune effector cell-associated cytotoxicity syndrome (ICANS). Cytokine release syndrome results from T-cell over-stimulation likely arising from the high affinity engagement of CD3 [84,85]. Regardless of the subclass of TCE, approximately half of patients develops CRS of any grade during cycle one (mosunetuzumab: 23%, glofitamab: 50,3%, odronextamab: 57%, epcoritamab: 57%, blinatumomab: 44%) and only minimum during the next cycles [25,67,73,76]. Stepwise dose escalation, corticosteroid premedication as well as tocilizumab (anti-IL-6 mAb) administration in the early onset of CRS are effective and significantly reduce the number of grade ≥3 AEs [86]. It is still unclear whether ICANS is attributed to on-target toxicity or more likely to damage caused by excessive CRS with hyperinflammation and disruption of the blood-brain barrier [81,87]. Therefore, efforts to mitigate CRS may indirectly reduce the incidence of associated neurotoxicity. Undoubtedly, except blinatumomab which required treatment discontinuation in several studies [73,76], severe ICANS occur rarely, and frequency of grade 1 or 2 differs among TCEs (ICANS of any grade reported in mosunetuzumab: 45%, glofitamab: 5%, epcoritamab: 6%, blinatumomab 9–56%) [26,65,69,76]. Due to the risk of IL-6 increase in central nervous system after tocilizumab administration, corticosteroids are recommended for the treatment of ICANS while combination with tocilizumab is recommended for CRS and concurrent ICANS [86]. Other common severe AEs were hematologic toxicity, dominantly neutropenia, and lymphopenia along with infections [27,65,76]. Subcutaneous application reduces the number of severe CRS. However, more local injection-site reactions were reported (47%) [26].

## 5. Conclusions

From huge amount of ADCs, loncastuximab tesirine, anti-CD19 ADC, is the only one that has been approved for RR DLBCL as a single agent [34]. Other ADCs used for aggressive B-NHL subsets work more effectively in combination with standard chemoimmunotherapy (e.g., PV plus R-CHP, BV plus R-CHP, BV plus BR) [19,31,52]. Furthermore, long-lasting follow-up is being awaited for chemotherapy-free regimens such as BV plus nivolumab in RR PMBCL [88], triplet of PV plus rituximab and lenalidomide in RR DLBCL [14], PV plus obinutuzumab and lenalidomide [15] or PV plus Ven [32] in RR FL.

Even though bsAbs, namely blinatumomab, has been tested in clinical trials for B-NHL since 2011, TCEs are still quite young molecules with no one commonly used in clinical practice for B-cell lymphomas and CLL treatment. Three anti-CD20xCD3 TCEs (epcoritamab, mosunetuzumab, glofitamab) have already entered phase 3 clinical trials (NCT04628494, NCT04712097, NCT04408638). The most promising seems to be epcoritamab, showing high response rates in both indolent and aggressive pretreated lymphoma patients [26]. Further investigation will reveal the potential of combinatory regimens such as glofitamab plus obinutuzumab [25], or mosenutuzumab plus PV [23].

In terms of anti-CD19 immunotherapeutic modalities, blinatumomab, loncastuximab tesirine, along with coltuximab ravtansine, are the only ADCs and TCEs clinically evaluating in B-NHL patients. This is quite the opposite to anti-CD19 CAR-T cells that have been widely used and gradually approved for aggressive and indolent B-NHL subtypes [89,90,91]. Furthermore, tafasitamab-cxix (*MorphoSys US*), which is a humanized Fc-modified cytolytic CD19 targeting mAb, has been approved by FDA in combination with lenalidomide for the treatment of RR DLBCL patient. It will be interesting to observe the development of anti-CD19 TCEs and ADCs in this wide field of immunotherapeutic agents, as they also have the advantage of so-called “off the shelf” products.

Despite the fact that anti-CD20xCD3 TCEs have been highly effective in indolent lymphomas, only limited data are available for TCEs used in CLL. At the moment, we can only speculate if this is because of primary impaired T-cell function [89,92,93] predicting failure of TCEs, too small CLL cohorts in clinical trials, or we just have to wait for studies with a dual or triple combination of TCE, Bcl-2 inhibitor, and/or B-cell receptor inhibitor.

## Figures and Tables

**Figure 1 ijms-22-11470-f001:**
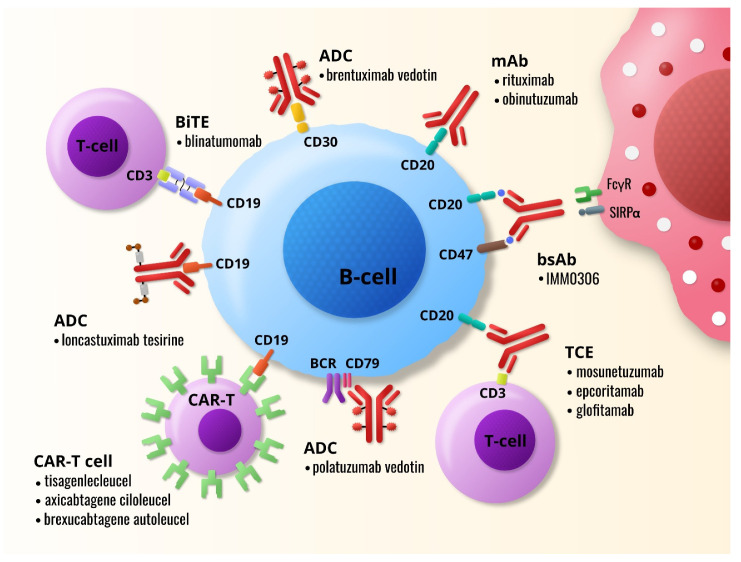
Examples of currently used and tested immunotherapeutic modalities in the treatment of B-cell non-Hodgkin lymphomas. mAb, monoclonal antibody; bsAb, bispecific antibody; TCE, T-cell engager; BiTE, bispecific T-cell engager; CAR-T cell, chimeric antigen receptor T cell; ADC, antibody-drug conjugate; BCR, B-cell receptor.

**Table 1 ijms-22-11470-t001:** Antibody-drug conjugates approved by FDA for hematooncological and oncological disorders.

ADC	Diagnosis
gemtuzumab ozogamicine (Mylotarg, Pfizer)	acute myeloid leukemia
brentuximab vedotin (Adcetris, Seagen/Takeda Oncology)	classical Hodgkin lymphoma specific CD30+ T-cell lymphomas
inotuzumab ozogamicin (Besponsa, Pfizer)	acute lymphoblastic leukemia
polatuzumab vedotin (Polivy, Roche)	diffuse large B-cell lymphoma
belantamab mafodotin (Blenrep, GlaxoSmithKline)	multiple myeloma
loncastuximab tesirine (Zynlonta, ADC Therapeutics S.A.)	large B-cell lymphomas
moxetumomab pasudotox (Lumoxiti, Astrazeneca)	hairy cell leukemia
trastuzumab emtansine (Kadcyla, Genentech, Roche)	HER2-positive metastatic breast cancer
trastuzumab deruxtecan (Enhertu, AstraZeneca/Daiichi Sankyo)	HER2-positive breast cancer
sacituzumab govitecan (Trodelvy, Immunomedics)	triple-negative breast cancer
enfortumab vedotin (Padcev, Astellas/Seattle Genetics)	urothelial cancer

**Table 2 ijms-22-11470-t002:** Antibody-drug conjugates-available results of phase I-III clinical trials; EN, estimated enrollment; ORR, overall response rate; CR, complete remission; PR, partial remission; NA, not available; mPF, median progression free survival; mOS, median overall survival; m, month; RR, relapse and/or refractory; Ph, phase; B-NHL, B-cell non-Hodgkin lymphoma; I, indolent; a, aggressive; N, number; DLBCL, Diffuse large B-cell lymphoma; FL, follicular lymphoma; GZL, gray zone lymphoma; PV, polatuzumab vedotin; BV, brentuximab vedotin; Lonca, Loncastuximab tesirine; CHOP, cyclophosphamide, doxorubicin, vincristine, prednisone; CHP, cyclophosphamide, doxorubicin, prednisone; R, rituximab; G, Obinutuzumab; BR, bendamustine, rituximab; BG, bendamustin, obinutuzumab; NR, not reached; NE, not evaluable; m, months.

Title	Diagnosis	EN	Regimen	ORR % (N)	CR % (N)	mPFS	mOS	Ph	Identifier Reference Status
POLATUZUMAB VEDOTIN									
A Study of Polatuzumab Vedotin (DCDS4501A) in Combination With Rituximab or Obinutuzumab Plus Bendamustine in Participants With Relapsed or Refractory Follicular or Diffuse Large B-Cell Lymphoma	RR DLBCL RR FL	331	Arm 1: PV + BR Arm 2: PV + BG Arm 3: BR	45% (18/40) 41% (11/27) 18% (7/40)	40% (16/40) 30% (8/27) 18% (7/40)	7.6m 6.3m 2.0 m	12.4m 10.8m 4.7 m	I/II	NCT02257567 [12] ongoing, not recruiting
A Study of Polatuzumab Vedotin in Combination With Rituximab or Obinutuzumab, Cyclophosphamide, Doxorubicin, and Prednisone in Participants With B-Cell Non-Hodgkin’s Lymphoma	RR B-NHL	90	Arm 1: PV + R-CHP Arm 2: PV + G-CHP	89% (40/45) 90% (19/21)	76% (34/45) 81% (17/21)	NE	NE	I/II	NCT01992653 [13] completed
A Study of Obinutuzumab, Polatuzumab Vedotin, and Lenalidomide in Relapsed or Refractory Follicular Lymphoma (FL) and Rituximab in Combination With Polatuzumab Vedotin and Lenalidomide in Relapsed or Refractory Diffuse Large B-Cell Lymphoma (DLBCL)	RR DLBCL	116	PV+R+Len	74% (36/49)	35% (17/49)	6.3m NE	10.9m NA	I	NCT02600897 [14] [15] ongoing, not recruiting
	RR FL		PV+G+Len	83% (38/46)	61% (28/46)				
LONCASTUXIMAB TESIRINE									
Study to Evaluate the Efficacy and Safety of Loncastuximab Tesirine in Patients With Relapsed or Refractory Diffuse Large B-Cell Lymphoma (LOTIS-2)	RR DLBCL	145	Lonca	48.3% (70/145)	24.8% (36/145)	4.9m	9.5m	II	NCT03589469 [16] ongoing, not recruiting
Study of ADCT-402 in Patients With Relapsed or Refractory B-cell Lineage Non Hodgkin Lymphoma (B-NHL)	RR B-NHL	183	Lonca	45.6% (82/180)	26.7 (48/180)	3.1 m	8.3 m	I	NCT02669017 [17] Completed
Safety and Efficacy Study of Loncastuximab Tesirine + Ibrutinib in Diffuse Large B-Cell or Mantle Cell Lymphoma	RR DLBCL RR MCL	161	Lonca + ibrutinib	63.9% (23/36)	36.1% (13/36)	NA	NA	I/II	NCT03684694 [18] ongoing
BRENTUXIMAB VEDOTIN									
An Investigational Immuno-therapy Safety and Effectiveness Study of Nivolumab in Combination With Brentuximab Vedotin to Treat Non-Hodgkin Lymphomas (CheckMate 436)	RR DLBCL RR PMBCL	146	BV + nivolumab	73% (22/30)	37% (11/30)	NA	NR	I/II	NCT02581631 [16] ongoing, not recruiting
Brentuximab Vedotin and Chemotherapy in CD30+ PMBL, Diffuse Large B-Cell, and Grey Zone Lymphoma Patients	ND DLBCL ND PMBCL ND GZL	32	BV + R-CHP	100% (29/29)	86% (25/29)	NA	NA	I/II	NCT01994850 [19] completed

**Table 3 ijms-22-11470-t003:** Antibody-drug conjugates-phase III ongoing clinical trials and phase I-II ongoing chemotherapy free clinical trials with no available results (NCT03677154, NCT03533283, NCT03671018 are included in Table 4; EN, estimated enrollment; RR, relapse and/or refractory; ND, newly diagnosed; B-NHL, B-cell non-Hodgkin lymphoma; DLBCL, Diffuse large B-cell lymphoma; FL, follicular lymphoma; PV, polatuzumab vedotin; CHOP, cyclophosphamide, doxorubicin, vincristine, prednisone; miniCHOP, dose modified CHOP, CHP, cyclophosphamide, doxorubicin, prednisone; miniCHP, dose modified CHP; R, rituximab; Gem, gemcitabine; Ox, oxaliplatin; Len, lenalidomide; ICE, ifosfamide, carboplatin, etoposide; PV, polatuzumab vedotin; BV, brentuximab vedotin; Lonca, loncastuximab tesirine; BR, bendamustine, rituximab.

ADC	Diagnosis	EN	Regimen	Phase	Identifier	Status
PV	RR DLBCL	216	Arm 1: PV+R-GemOx Arm 2: R-GemOx	III	NCT04182204	ongoing
PV	RR DLBCL	42	Arm 1: PV + BR Arm 2: BR	III	NCT04236141	ongoing, not recruiting
PV	ND DLBCL	1000	Arm 1: PV + R-CHP Arm 2: R-CHOP	III	NCT03274492	ongoing, not recruiting
PV	ND DLBCL ND FL	200	Arm 1: PV + R-miniCHP Arm 2: PV + R-miniCHOP	III	NCT04332822	ongoing
PV	RR DLBCL	334	Arm 1: PV + R-ICE Arm 2: R-ICE	III	NCT04833114	ongoing
Lonca	RR DLBCL	350	Arm 1: Lonca-R Arm 2: R-GemOx	III	NCT04384484	ongoing
BV	RR DLBCL	400	Arm 1: BV+R+Len Arm 2: R+Len	III	NCT04404283	ongoing
PV	RR MCL	63	PV + R+Ven+Hyalur + Ven+Hyalur©	I/II	NCT04659044	not yet recruiting
PV	RR B-NHL	252	Arm 1: PV + CC-220 + R Arm 2: CC-220 + Tafasitamab Arm 3: CC-220 + R-GDP	I/II	NCT04882163	not yet recruiting
PV	RR FL	133	PV+Obi+Ven + Obi+Venµ	I	NCT02611323	ongoing
	RR DLBCL		PV+R+Ven + R+Ven*			
Lonca	RR FL	150	Lonca + idelalisib	I	NCT04699461	ongoing

**Table 4 ijms-22-11470-t004:** Anti-CD20xCD3 TCEs-available results of phase I-II clinical trials; EN, estimated enrollment; ORR, overall response rate; CR, complete remission; PR, partial remission; mPF, median progression free survival; mOS, median overall survival; m, month; RR, relapse and/or refractory; B-NHL, B-cell non-Hodgkin lymphoma; i, indolent; a, aggressive; N, number; NA, not available; Obi, obinutuzumab; Ph, phase; * obinutuzumab pretreatment; # subcutaneous application.

Title	Diagnosis	EN	Regimen	ORR % (N)	CR % (N)	mPFS	mOS	Ph	Identifier Reference Status
MOSUNETUZUMAB									
A Safety and Pharmacokinetic Study of BTCT4465A (Mosunetuzumab) as a Single Agent and Combined With Atezolizumab in Non-Hodgkin’s Lymphoma and Chronic Lymphocytic Leukemia (CLL)	RR B-NHLi RR B-NHLa ND DLBCL	836	Arm 1: mosunetuzumab	68% (42/62) 34.7%(41/119) 67.7% (21/31)	50% (31/62) 18.6% (22/119) 41.9%(13/31)	11m	NA	I/II	NCT02500407 [20] [21] [22] ongoing
	RR B-NHLi RR B-NHLa		Arm 2: mosunetuzumab #	86% (6/7) 60% (9/15)	29% (2/7) 20% (3/15)	NA	NA		
A Study to Evaluate the Safety and Efficacy of Mosunetuzumab (BTCT4465A) in Combination With Polatuzumab Vedotin in B-Cell Non-Hodgkin Lymphoma	RR B-NHL	262	Arm 1: mosunetuzumab + PV	68% (15/22)	54.5% (12/22)	NA	NA	I/II	NCT03671018 [23] ongoing
GLOFITAMAB									
A Dose Escalation Study of Glofitamab (RO7082859) as a Single Agent and in Combination With Obinutuzumab, Administered After a Fixed, Single Pre-Treatment Dose of Obinutuzumab in Participants With Relapsed/Refractory B-Cell Non-Hodgkin’s Lymphoma	RR B-NHLa RR B-NHLi	860	Arm 1: Obi * + glofitamab	48.0% (61/127) 70.5% (31/44)	33.1% (42/127) 47.7% (21/44)	2.9m 11.8m	NA NA	I/II	NCT03075696 [24] [25] ongoing
	RR B-NHLa RR FL		Arm 2: Obi * + glofitamab + Obi	38% (6/16) 80% (4/5)	31% (5/16) 80% (4/5)	NA	NA		
EPCORITAMAB									
GEN3013 Trial in Patients With Relapsed, Progressive or Refractory B-Cell Lymphoma	RR DLBCL RR FL	486	epcoritamab #	91% (10/11) 80% (10/12)	55% (6/11) 60% (7/12)	NA	NA	I/II	NCT03625037 [26] ongoing
ODRONEXTAMAB									
Study to Investigate the Safety and Tolerability of Odronextamab in Patients With CD20+ B-Cell Malignancies (ELM-1)	RR B-NHLa RR B-NHLi RR CLL	256	odronextamab	33% (15/45) 93% (13/14) NA	18% (8/45) 71.4% (10/14) NA	NA NA NA	NA NA NA	I	NCT03888105 [27] ongoing

**Table 5 ijms-22-11470-t005:** Bispecific antibodies-phase I-III ongoing clinical trials with no available results; bsAb, bispecific antibody; EN, estimated enrollment; RR, relapse and/or refractory; ND, newly diagnosed; B-NHL, B-cell non-Hodgkin lymphoma; DLBCL, Diffuse large B-cell lymphoma; HGBCL, high-grade B-cell lymphoma; MCL, mantle cell lymphoma; MZL, marginal zone lymphoma; CLL, chronic lymphocytic leukemia; FL, follicular lymphoma; tFL, transformed follicular lymphoma; atezo, atezolizumab; PV, polatuzumab vedotin; CHOP, cyclophosphamide, doxorubicin, vincristine, prednisone; CHP, cyclophosphamide, doxorubicin, prednisone; R, rituximab; mosunetuzumab, TCE anti-CD20xCD3, glofitamab, TCE anti-CD20xCD3; RO7227166, TCE anti-CD19x4-1BB; plamotamab, TCE anti-CD20xCD3; IGM-2323, TCE anti-CD20xCD3; GB261, TCE anti-CD20xCD3; Gem gemcitabine; Ox, oxaliplatin; Obi, obinutuzumab; Len, lenalidomide; * obinutuzumab pretreatment; § first results from arm 1 are presented in Table 1; # subcutaneous application.

bsAb	Diagnosis	EN	Regimen	Phase	Identifier	Status
mosunetuzumab	RR FL	400	Arm 1: mosunetuzumab + Len Arm 2: rituximab + Len	III	NCT04712097	not yet recruiting
glofitamab	RR DLBCL	270	Arm 1: Obi * + glofitamab + GemOx Arm 2: R-GemOx	III	NCT04408638	ongoing
epcoritamab	RR DLBCL RR tFL		Arm 1: epcoritamab # Arm 2: R-GemOx or BR	III	NCT04628494	ongoing
mosunetuzumab	ND FL ND MZL	52	mosunetuzumab # + Len	II	NCT04792502	not yet recruiting
mosunetuzumab glofitamab	RR B-NHLa RR tFL	42	Arm 1: mosunetuzumab Arm 2: glofitamab + Obi	II	NCT04889716	not yet recruiting
glofitamab	RR B-NHL	78	Obi * + glofitamab	II	NCT04703686	ongoing
odronextamab	RR B-NHL	512	odronextamab	II	NCT03888105	ongoing
mosunetuzumab	ND DLBCL	40	Arm 1: mosunetuzumab Arm 2: mosunetuzumab # + PV	I/II	NCT03677154	ongoing
mosunetuzumab	RR MCL	262	Arm 2: mosunetuzumab # + PV	I/II	NCT03671018	ongoing
mosunetuzumab	ND DLBCL RR B-NHL	160	Arm 1: mosunetuzumab + PV + CHP Arm 2: mosunetuzumab + CHOP	I/II	NCT03677141	ongoing, not recruiting
glofitamab	RR B-NHL	140	Arm 1: Obi * + glofitamab + atezo Arm 2: Obi *+ glofitamab + PV	I/II	NCT03533283	ongoing
glofitamab	ND DLBCL ND HGBCL	80	Arm 1: glofitamab + R-CHOP Arm 2: glofitamab + R-CHOP + PV-CHOP	I/II	NCT04914741	not yet recruiting
epcoritamab	RR CLL	32	epcoritamab	I/II	NCT04623541	ongoing
epcoritamab	RR DLBCL RR DLBCL ND DLBCL RR FL ND FL	130	Arm 1: epcoritamab + R-DHAX/C Arm 2: epcoritamab + GemOx Arm 3: epcoritamab + R-CHOP Arm 4: epcoritamab + R-Len Arm 5: epcoritamab + BR	I/II	NCT04663347	ongoing
epcoritamab	RR B-NHL	73	epcoritamab #	I/II	NCT04542824	ongoing
GB261	RR B-NHL RR CLL	460	GB261	I/II	NCT04923048	not yet recruiting
mosunetuzumab	RR B-NHL RR CLL	836	Arm 1: mosunetuzumab § Arm 2: mosunetuzumab + atezo	I	NCT02500407	ongoing
mosunetuzumab glofitamab	RR FL	27	Arm 1: mosunetuzumab + Len Arm 2: mosunetuzumab # + Len Arm 3: Obi * + glofitamab + Len Arm 4: Obi * + glofitamab + Len + Obi	I	NCT04246086	ongoing
mosunetuzumab glofitamab	RR DLBCL RR HGBCL	20	Arm 1: Obi * + glofitamab + Gem + Ox Arm 2: mosunetuzumab + Gem + Ox	I	NCT04313608	ongoing, not recruiting
glofitamab	RR DLBCL	30	Arm 1: glofitamab Arm 2: Obi * + glofitamab	I	NCT04657302	ongoing
glofitamab	RR B-NHL	207	Arm 1: Obi * + glofitamab + RO7227166 Arm 2: Obi * + glofitamab + Obi	I	NCT04077723	ongoing
glofitamab	RR B-NHL ND DLBCL	172	Arm 1: glofitamab + R-CHOP Arm 2: glofitamab + Obi-CHOP	I	NCT03467373	ongoing
epcoritamab	RR B-NHL	486	epcoritamab #	I	NCT03625037	ongoing
odronextamab	RR B-NHL RR HL	172	cemiplimab + odronextamab	I	NCT02651662	not yet recruiting
plamotamab	RR B-NHL RR CLL	216	plamotamab	I	NCT02924402	ongoing
IGM-2323	RR B-NHL	160	IGM-2323	I	NCT04082936	ongoing

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
