# Peer review of "Promising Immunotherapeutic Modalities for B-Cell Lymphoproliferative Disorders"

_ijms, 2021, doi:10.3390/ijms222111470_

Round 1

Reviewer 1 Report

The review article of Mihályová J et al on the role of antibody-drug conjugates (ADCs) and bispecific antibodies (bsAbs) in B-NHL and CLL treatment is interesting because it deals with one of the most current challenges in cancer therapy.

Some changes are important to improve the work:

  • Page 3: transform the last part of paragraph Antibody-drug conjugates (from "As of 2021 onwards ...) into a table listing the nine approved ADCs.
  • Page 6: move the sentence that mentions tables 1 and 2 (Results of phase I/II clinical…) to the end of paragraph 2, allowing the reader to progressively visualize in the table what is described in the text.
  • Page 6: I would suggest that the authors add a figure in the paragraph 4 to graphically illustrate the different formats of bsAbs.
  • Page 9: as for the previous comment on tables 1 and 2, move the sentence that mentions tables 3 and 4 to the beginning of paragraph 4.1
  • Page 9: Add in the references section the citation of Webster et al 2019, present in the text.
  • Renumber the paragraphs: 3.1 is 2.1, 3.1.1 is 2.1.1 and so on.
  • Please correct typographical errors throughout the text.

Reviewer 2 Report

The Authors provide a narrative review on the role of antibody-drug conjugates and bispecific antibodies in the treatment of B-cell NHL and CLL, focusing on mechanism of action, treatment-related toxicities, pre-clinical results and clinical trials. At first glance, this is a well-written reviewe article, with a clear outline and a straightforward informative path. The authors should be commended for this effort. Few minor comments:

  • Please clearly identify and state: population, intervention, comparator, outcome, timing and setting (PICOTS).
  • Please provide some information for the search strategy that has been employed for the present review article.
